# Factors Affecting the Progression of Infection-Related Glomerulonephritis to Chronic Kidney Disease

**DOI:** 10.3390/ijms22020905

**Published:** 2021-01-18

**Authors:** Takashi Oda, Nobuyuki Yoshizawa

**Affiliations:** 1Kidney Disease Center, Department of Nephrology and Blood Purification, Tokyo Medical University Hachioji Medical Center, Hachioji, Tokyo 193-0998, Japan; 2Hemodialysis Unit, Showanomori Hospital, Akishima, Tokyo 196-0024, Japan; yoshizawa@nomuramedical.com

**Keywords:** acute kidney injury, poststreptococcal acute glomerulonephritis, infection-related glomerulonephritis, chronic kidney disease, tubulo-interstitial change, nephritis-associated plasmin receptor, alpha-smooth muscle actin

## Abstract

Acute glomerulonephritis (AGN) triggered by infection is still one of the major causes of acute kidney injury. During the previous two decades, there has been a major paradigm shift in the epidemiology of AGN. The incidence of poststreptococcal acute glomerulonephritis (PSAGN), which develops after the cure of group A *Streptococcus* infection in children has decreased, whereas adult AGN cases have been increasing, and those associated with nonstreptococcal infections, particularly infections by *Staphylococcus*, are now as common as PSAGN. In adult AGN patients, particularly older patients with comorbidities, infections are usually ongoing at the time when glomerulonephritis is diagnosed; thus, the term “infection-related glomerulonephritis (IRGN)” has recently been popularly used instead of “post-infectious AGN”. The prognosis of children with PSAGN is generally considered excellent compared with that of adult IRGN cases. However, long-term epidemiological analysis demonstrated that an episode of PSAGN in childhood is a strong risk factor for chronic kidney disease (CKD), even after the complete remission of PSAGN. Although the precise mechanism of the transition from IRGN to CKD remains unknown, its clarification is important as it will lead to the prevention of CKD. In this review, we therefore focus on the possible factors that may contribute to the progression of IRGN into CKD. Four factors, namely, persistent infection, genetic background of the host’s complement system, tubulointerstitial changes, and pre-existing histological damage, are discussed.

## 1. Introduction

Acute kidney injury (AKI) has been increasing in the previous few decades and has recently been recognized as an important cause of chronic kidney disease (CKD), which may progress to end stage renal disease (ESRD) [1]. However, the precise mechanism of the transition from AKI to CKD remains obscure and is a matter of great concern.

Acute glomerulonephritis (AGN) triggered by infection is still one of the major causes of AKI. During the past century, poststreptococcal acute glomerulonephritis (PSAGN) that develops after the cure of group A *Streptococcus* (GAS) infection after a distinct latent period in children comprised the majority of AGN cases [2,3]. However, during the previous two decades, there has been a major paradigm shift in the epidemiology of AGN. Probably owing to the improvement in living environments and the adequate use of antibiotics, the incidence of PSAGN has decreased, particularly in developed countries. On the other hand, adult AGN cases have been increasing, and those associated with nonstreptococcal infections, particularly infection of *Staphylococcus*, are now as common as PSAGN. Furthermore, in adult AGN patients, particularly older patients with comorbidities, infections are usually ongoing at the time when glomerulonephritis is diagnosed. This is why the term “infection-related glomerulonephritis (IRGN)” has recently been more commonly used instead of “post-infectious AGN” [3]. Notably, whereas most PSAGN in children resolve without any specific treatment, the prognosis of adult IRGN is poor, and older patients, particularly those with immunocompromised backgrounds, such as diabetes mellitus, malignancies, or alcoholism, are reported to be at high risk [4].

Thus, typical PSAGN in children is considered as a benign disease with a favorable prognosis that completely resolves without progression, in contrast to IRGN in adults, which often progresses into chronicity with an unfavorable renal prognosis. However, a long-term epidemiological study demonstrated that an episode of PSAGN in childhood is a strong risk factor for CKD and ESRD in adulthood, even after the complete remission of PSAGN [5,6,7,8,9,10]. Although the precise mechanism of the transition from AGN to CKD remains unknown, understanding it is important as it is expected to lead to the prevention of CKD and ESRD.

In this review, we therefore focus primarily on the possible factors that may contribute to the progression of IRGN, which is a major cause of AKI, into CKD. As summarized in Table 1 and Figure 1, the following four factors are listed and discussed: 1. persistent infection, 2. genetic background of the host’s complement system, 3. tubulointerstitial changes, and 4. pre-existing histological damage due to old age and comorbidities. Among these factors, 2 of them (1 and 2) are associated with the pathogenic mechanism of IRGN, whereas the other 2 factors (3 and 4) are independent of IRGN itself.

Some autoantibodies, such as the anti-neutrophil cytoplasmic antibody (ANCA), anti-nuclear antibody (ANA), anti-dsDNA antibody, and anti-factor B antibody have been reported to be detected in patients with IRGN [14,15,16,17,18,19,20,21]. Although these antibodies may contribute to the progression of IRGN, at present there is little data regarding their significance on the prognosis of IRGN, and as their involvement remains controversial, we touched on this point but did not list the antibodies as possible factors associated with IRGN.

Generally, clear evidence in this field is very scarce because no large prospective clinical studies have been performed owing to the rarity of IRGN, and, furthermore, a reliable animal model has not been established to date, partly owing to the differences in the infectiveness of pathogens among different species. Therefore, a substantial portion of this review is based on detailed evaluations of case reports.

## 2. Persistent Infection as a Possible Cause of the Progression of IRGN into CKD

The simplest reason for the persistence and progression of IRGN into CKD is the persistence of the causative infection, resulting in the continuation of the pathogenic mechanism. In this setting, abnormalities in urinalysis, serum complement levels, and inflammation (ESR and CRP levels) continue, leading to disease progression into chronic glomerulonephritis. Many factors, such as the strain of pathogen (various bacteria and viruses), focus of infection, and conditions of the host (immune competence, comorbidities, use of indwelling devices, etc.) may affect the persistence of pathogens. In terms of bacterial strains, as described in the introduction, GAS infection tends to occur more frequently in children and is usually completely cured before the onset of glomerulonephritis. Glomerular histological analysis in such a condition usually shows the so-called acute change, i.e., prominent endocapillary proliferation mainly by the accumulation of infiltrating cells [22,23]. On the other hand, *Staphylococcal* IRGN mainly affects older adults who often have comorbidities, and the infection is ongoing when the glomerulonephritis develops [4]. Glomerular histological changes in such patients with ongoing infection may also show endocapillary proliferative glomerulonephritis in the early phase of the disease course. However, as the disease duration after the onset of glomerulonephritis becomes longer, glomerular changes appear to make a gradual transition from endocapillary proliferative glomerulonephritis to membranoproliferative glomerulonephritis (MPGN) or mesangial proliferative glomerulonephritis (MesPGN), probably through chronic glomerular endothelial damage and gradual transition from the accumulation of infiltrating cells to the proliferation of mesangial cells. *Staphylococcal* infections in older patients, particularly deep-seated infections, are frequently occult in nature and are quite difficult to detect. Therefore, complicating glomerulonephritis tends to be detected in its chronic stages, and histological analysis often shows a MPGN or MesPGN pattern with or without crescent formation and IgA deposition. Typical examples of this condition have been reported in IRGN caused by infective endocarditis due to *Staphylococcus aureus* and caused by ventriculoatrial shunt infections due to coagulase-negative *Staphylococcus epidermidis*.

For assessment of the continuation of the pathogenic mechanism due to persistent infection, the identification of histological biomarkers is desired. In this respect, nephritis-associated plasmin receptor (NAPlr) and associated plasmin activity may be useful [11,24,25]. NAPlr was originally isolated from the cytoplasmic fraction of GAS as a candidate nephritogenic protein of PSAGN, and was found to be the same molecule as streptococcal glyceraldehyde-3-phosphate dehydrogenase (GAPDH) [24,25]. Glomerular NAPlr deposition is frequently observed by immunofluorescence staining in early stage PSAGN patients (Figure 2); all patients within two weeks of disease onset are reported to show NAPlr deposition [25]. The deposited NAPlr binds with plasmin and maintains its activity by protecting it from physiological inhibitors, and is considered to cause glomerular damage directly by degrading extracellular matrix proteins and indirectly by activating pro-matrix metalloproteases. Additionally, glomerular plasmin activity can exert proinflammatory functions by activating and accumulating inflammatory cells [11].

Recently, glomerular NAPlr deposition and associated plasmin activity were reported to be observed not only in patients with PSAGN but also in those with other glomerular diseases, in whom preceding streptococcal infection had been suggested [26,27,28,29,30,31]. In fact, the preceding infection might be an infection other than GAS, because the GAPDH of various bacteria show cross-immunoreactivity to the anti-NAPlr antibody, and simultaneously show plasmin-binding function [32,33,34]. From these results, NAPlr and associated plasmin activity are presently considered as general biomarkers of IRGN [35]. Positive glomerular staining of these markers usually disappears within 30 days after the onset of PSAGN [25]. However, the prolonged positive glomerular staining of these markers (for more than half a year) has been observed in some IRGN patients, suggesting persistent infection and its pathogenic significance in these patients [26,27].

It is very important to shed light on the possible involvement of persistent infection in the pathogenic condition of glomerulonephritis, because this factor is potentially modifiable. Using NAPlr and plasmin activity as biomarkers, the pathogenic involvement of persistent infection can be detected. If these biomarkers are persistently positive, the most important therapeutic strategy would be to eradicate the persistent and pathogenic infection, which may result in blocking the transition of IRGN to CKD. Indeed, Noda et al. recently reported an interesting IRGN case caused by asymptomatic sinusitis, which suggests the importance of detecting the hidden infection by histological staining of NAPlr and plasmin activity. Eradication of the hidden but pathogenic infection in this patient resulted in clinical remission of the disease (Figure 3) [36].

The histological transition from endocapillary proliferative glomerulonephritis to MPGN is also observed in some cases of viral IRGN. Indeed, we encountered a patient in which the first renal biopsy showed endocapillary proliferative glomerulonephritis typical of AGN associated with parvovirus B19 (PVB19) infection, and in the second biopsy, which was performed 4 years subsequently because of persistent proteinuria and prolonged low serum complement C3 level with positivity for the IgM antibody for PVB19 (persistent PVB19 infection), showed MPGN with mesangial interposition and with thickening and double contours of the glomerular basement membrane (GBM) [37]. This case provides lines of evidence that the transition from acute endocapillary proliferative glomerulonephritis to MPGN can actually occur during prolonged infection.

## 3. Genetic Background of the Host’s Complement Regulatory System as a Possible Cause of the Progression of IRGN into CKD

There have been several reports of patients who have acute nephritic syndrome with preceding streptococcal infection, diagnosed initially as having PSAGN, but owing to the persistence of hypocomplementemia and urinary abnormalities, genetic testing of the alternative complement pathway was performed, and a genetic abnormality in the complement regulatory system was eventually identified, leading to the final diagnosis of C3 glomerulopathy [38,39]. These cases suggest that in the presence of an abnormal genetic background of the host’s complement system, the pathogenic mechanism of glomerulonephritis associated with complement activation induced by infection continues, which may turn PSAGN into persistent smoldering nephritis causing C3 glomerulopathy, leading to the development of CKD without resolution. As the disease entity and the diagnostic criteria for both diseases (IRGN and C3 glomerulopathy) are not yet fully established, patients with a similar condition have also been reported as having atypical post-infectious glomerulonephritis (PIGN) [12]. In this previous report, 11 patients with so-called atypical PIGN were analyzed and 4 of them were found to have genetic abnormalities in the regulatory system of the alternative complement pathway (3 had a mutation in CFH, and 1 in CFHR5). In this setting, the pathogenic mechanism associated with complement activation inducing glomerulonephritis persists not because of persistent infection but because of the host’s characteristic genetic background regarding the complement regulatory system, and abnormalities in urinalysis and low serum complement levels continue without normalization, leading to the subsequent progression into chronic glomerulonephritis.

Considering the rapid recent advances in the development of medications that control complement activation, such as eculizumab and avacopan, this factor may also be potentially modifiable in the future, using appropriate complement-regulating medications.

## 4. Continuous Tubulointerstitial Changes Independent of Glomerular Changes as a Possible Cause of the Progression of IRGN into CKD

In 1961, Jennings and Earle [40] reported that the degree of interstitial reaction is one of the main factors determining whether PSAGN becomes chronic or not. Similar findings have actually been reported more generally, i.e., the progression of glomerular disease correlates more closely with interstitial lesions than with glomerular lesions [41,42,43,44].

In association with these observations, there have been reports indicating that an increase in the number of interstitial alpha-smooth muscle actin (α-SMA)-positive cells, which are believed to be myofibroblasts, is a good marker of disease progression in various types of human glomerulonephritis, including IgA nephropathy [45], membranous nephropathy [46], and diabetic nephropathy [47].

We therefore analyzed the α-SMA expression immunohistochemically, using a monoclonal antibody against α-SMA and renal biopsy tissues sampled repeated from 6 PSAGN patients, 3 of whom progressed into CKD, and 3 who were cured. Glomerular mesangial cells have been reported to show an α-SMA-positive phenotype, with cell activation or proliferation in both experimental and human glomerulonephritis [48].

Consistent with the report, prominent glomerular α-SMA expression was observed in all six tissues of patients in the acute phase of PSAGN, irrespective of disease prognosis, whereas glomerular α-SMA expression was uniformly minimal or absent in tissues of patients in the convalescent phase (Table 2, Figure 4). Namely, there was no significant difference in glomerular α-SMA expression between those who progressed into CKD and those who were cured, both in the acute phase and in the convalescent phase. Interstitial α-SMA expression was also generally upregulated in acute phase tissues, but the expression was more prominent in tissues of patients who progressed into CKD than in the cured patients, and this tendency persisted in the convalescent phase. These data indicate that glomerular α-SMA expression appears to be a reversible and transient phenomenon, and levels return to the steady state irrespective of the disease prognosis. In contrast, interstitial α-SMA expression appears to be associated with the development of chronic disease.

Thus, from the view of long-term renal prognosis, it is obvious that tubulointerstitial changes are more important than glomerular changes in PSAGN. However, although PSAGN is clearly a glomerular disease, the precise mechanism as to how tubulointerstitial changes develop in PSAGN has not been fully elucidated. Regarding this point, results from our previous study on proliferating cells in PSAGN tissues is informative [23]. In this previous study, we found a prominent increase in proliferating cell nuclear antigen (PCNA)-positive cells in kidney tissues of PSAGN patients, but the distribution of the PCNA-positive cells was irregular (Figure 5). This suggested that in PSAGN, locally produced growth factors, rather than circulating factors, act on the glomerular cells and subsequently the adjacent tubular epithelial cells of the nephron. This means that specific factors produced in the glomeruli filtrate into the urine and damage the tubular epithelial cells from the apical side. The damaged and activated PCNA-positive tubular epithelial cells may then affect the interstitial cells.

Alternatively, interstitial inflammation developing from the vascular pole of the glomeruli via the spread of massive glomerular inflammation may affect the adjacent tubular epithelial cells, inducing PCNA in the damaged tubules, as shown in Figure 6.

Tubulointerstitial changes that develop in this manner persist in the absence of glomerular inflammation, and may contribute to the progression of PSAGN.

The above findings are somewhat consistent with the recent reports on the importance of proximal tubular injury and its limited regenerative properties on the transition of AKI into CKD [49,50,51].

It may be possible to assess the involvement of this factor, tubulointerstitial changes, by the level of interstitial α-SMA expression; however, at present, there are no established therapeutic strategies that potentially modify this factor. The development of medication that modifies interstitial fibrosis, the final common pathway, are expected in the future.

## 5. Pre-Existing Kidney Injury as a Possible Cause of the Progression of IRGN into CKD

This factor represents adults with pre-existing comorbidities, such as nephrosclerosis or diabetic kidney disease, which may prevent them from achieving a complete recovery [4,13]. In fact, these cases should not be regarded as a transition from IRGN to CKD, because CKD actually exists before the onset of IRGN as a hidden condition. In addition, the onset of IRGN demonstrates the existing but hidden condition of CKD. As this factor is also potentially modifiable, doctors should focus on controlling the pre-existing comorbidities.

## 6. Detection of Autoantibodies in IRGN Patients

ANCA, especially PR3-ANCA, has been reported to be detected in patients with IRGN, particularly that caused by infective endocarditis [14,15,16,17]. There have also been reports of lupus-like symptoms in patients with some viral IRGN, such as parvovirus B19 [18,19] and CMV IRGN [20], with the appearance of ANA and anti-dsDNA antibodies. Furthermore, the presence of anti-factor B autoantibodies has recently been reported in patients with acute PIGN [21]. However, these were mostly reports of a small number of cases, and analyses of a large number of cases or prospective studies have not been reported to date. Furthermore, there appears to be a marked difference in outcomes depending on the report regarding the association between the appearance of autoantibodies and the prognosis of IRGN. Thus, the effect of the presence of autoantibodies on the prognosis of IRGN remains unclear, and is a matter of future investigation.

## 7. Concluding Remarks

IRGN is still considered to be one of the major causes of AKI, and the transition from IRGN to CKD has consistently been a focus of attention. In this review, we therefore summarized the possible factors contributing to the transition from IRGN to CKD.

Regarding the persistence of infection, positive glomerular staining for NAPlr and associated plasmin activity can be used as general histological markers. If these markers are persistently positive, eradication of the infection is the most important therapeutic strategy to stop the transition of IRGN to CKD.

Understanding the possible involvement of the genetic background of the host’s complement system and pre-existing comorbidities is also important, because both factors are potentially modifiable.

Regarding the tubulointerstitial changes, interstitial α-SMA staining was suggested to be useful for its assessment in IRGN patients. However, the precise mechanism underlying the association of glomerular damage and tubulointerstitial changes with α-SMA expression remains unknown. Furthermore, it is not known how tubulointerstitial change can be modified, and this is an important matter for future investigation.

## Figures and Tables

**Figure 1 ijms-22-00905-f001:**
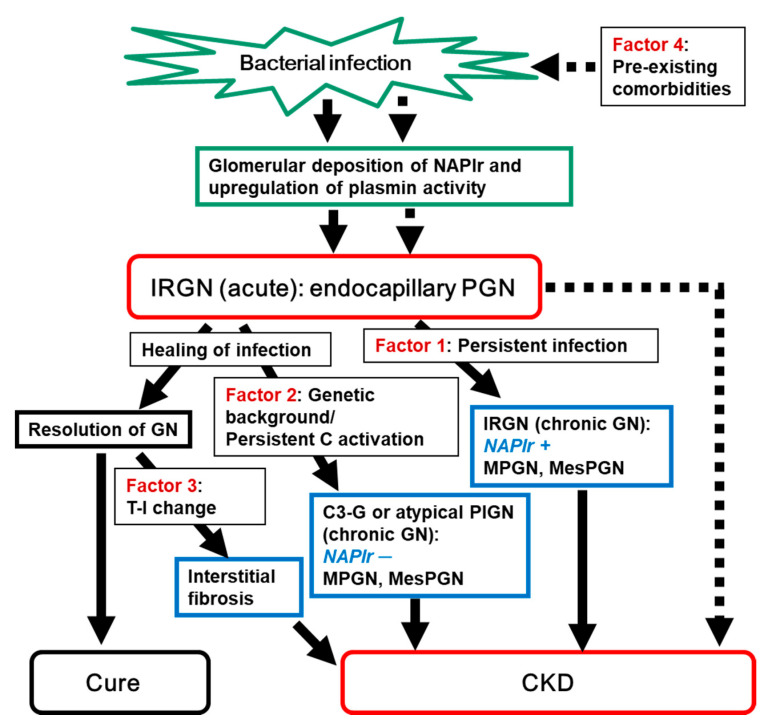
Summary of the concept of this review. Induction of IRGN and its outcome are summarized. IRGN may cure completely or may progress into CKD. Four factors that may contribute to the progression of IRGN into CKD are depicted. Solid arrow indicates the main flow of induction of IRGN and its outcome. While dotted arrow indicate the flow of patients with pre-existing comorbidities (Factor 4). NAPlr: nephritis-associated plasmin receptor; IRGN: infection-related glomerulonephritis; PGN: proliferative glomerulonephritis; GN: glomerulonephritis; C activation: complement activation; MPGN: membranoproliferative glomerulonephritis; MesPGN: mesangial proliferative glomerulonephritis; T-I: tubulo-interstitial; C3-G: C3 glomerulopathy; PIGN: postinfectious glomerulonephritis; CKD: chronic kidney disease.

**Figure 2 ijms-22-00905-f002:**
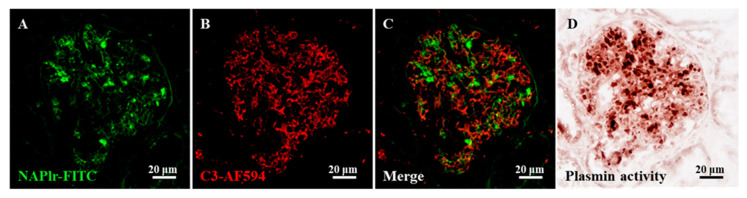
Representative photomicrographs of the histological staining for C3, nephritis-associated plasmin receptor (NAPlr), and plasmin activity in the glomeruli of a post-streptococcal acute glomerulonephritis (PSAGN) patient. (**A**–**C**) Double immunofluorescence (IF) staining for NAPlr (fluorescein isothiocyanate, green) and complement C3 (Alexa Fluor 594, red). Both NAPlr (**A**) and C3 (**B**) were positive in the glomeruli, but they generally were not colocalized, as shown in the merged image (**C**). (**D**) Plasmin activity assessed by in situ zymography on a serial section was found to be positive, and to have a similar distribution to the NAPlr staining in the glomeruli. Details of all staining methods have been described previously [11].

**Figure 3 ijms-22-00905-f003:**
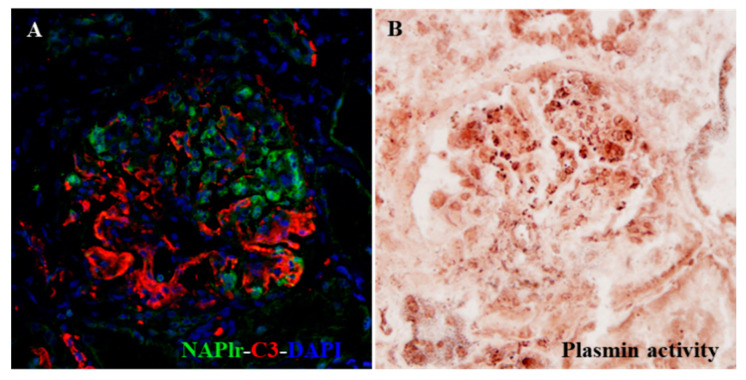
Photomicrographs of histological staining for C3, NAPlr, and plasmin activity in the glomeruli of a patient with infection-related glomerulonephritis (IRGN) induced by asymptomatic sinusitis [36]. Although the infection was clinically inapparent, double IF staining for NAPlr (fluorescein isothiocyanate, green) and C3 (Alexa Fluor 594, red) with nuclear staining for DAPI (blue) showed glomerular deposition of NAPlr and C3 (**A**). Furthermore, glomerular plasmin activity assessed by in situ zymography on a serial section demonstrated a similar distribution as NAPlr deposition, providing histological evidence for the substantial involvement of bacterial infection in the development of glomerulonephritis (**B**).

**Figure 4 ijms-22-00905-f004:**
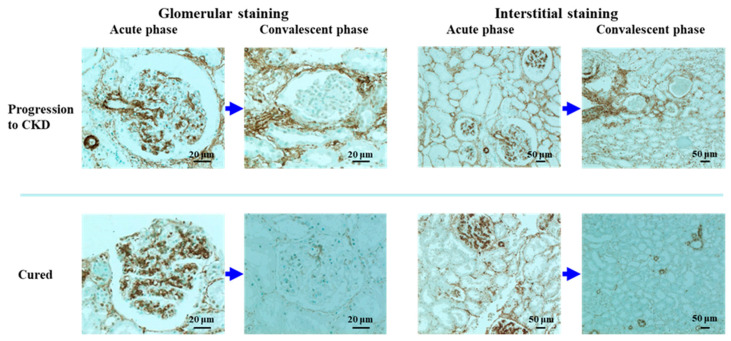
Immunoperoxidase staining for alpha-smooth muscle actin (α-SMA) with methyl green counterstaining in repeated renal biopsy sections from the acute phase and the convalescent phase of a PSAGN patient who developed chronic kidney disease (CKD), and a PSAGN patient who was cured. Strong positive staining of α-SMA in the mesangial area of the glomeruli was observed in tissues of PSAGN patients in the acute phase irrespective of disease prognosis, and this glomerular staining diminished uniformly in the convalescent phase irrespective of disease prognosis. Interstitial α-SMA staining was more intense in the PSAGN patient who developed CKD than in the patient who was cured, both in the acute phase and in the convalescent phase. For the indirect immunoperoxidase staining for α-SMA, a mouse monoclonal antibody (clone 1A4) was used as the primary antibody, and a peroxidase-conjugated goat anti-mouse IgG antibody was used as the secondary antibody.

**Figure 5 ijms-22-00905-f005:**
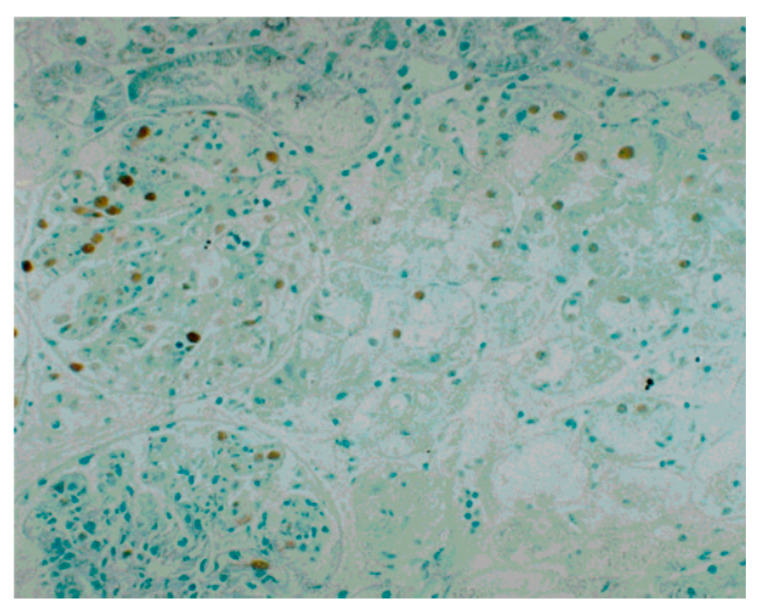
Immunoperoxidase staining for proliferating cell nuclear antigen (PCNA) and counterstaining with methyl green in a renal biopsy tissue from a PSAGN patient (original magnification, ×100) [23]. Prominent expression of PCNA in the glomerulus and tubular epithelial cells was observed. The distribution of PCNA-positive cells was variable; i.e., there was a considerable difference in the number of PCNA-positive cells between each glomerulus and between different parts of the tubules.

**Figure 6 ijms-22-00905-f006:**
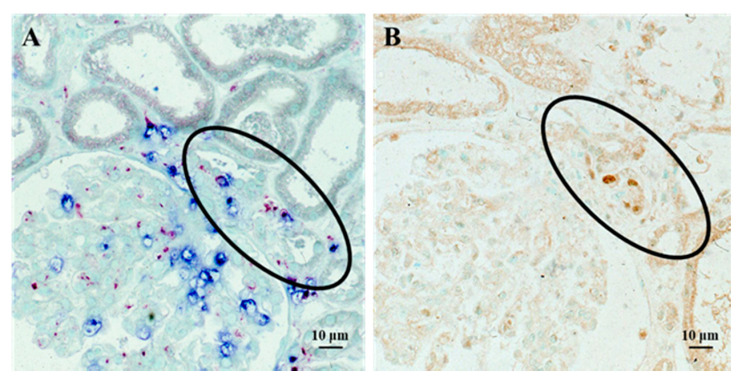
Staining of infiltrating cells and PCNA in serial sections of renal biopsy tissue from a patient with PSAGN. Double staining for neutrophils (chloroesterase staining: blue) and macrophages (immuno-alkaline phosphatase staining for CD11c: red) demonstrated extensive glomerular infiltration of both cell types, which spread out from the vascular pole (circled region) (**A**). Tubular epithelial cells of the same region (circled) showed positive PCNA staining (immunoperoxidase staining: brown) on a serial section (**B**). Details of all the staining methods have been described previously [22].

**Table 1 ijms-22-00905-t001:** Factors affecting the progression of infection-related glomerulonephritis to chronic kidney disease.

Factor	Evaluation of the Involvement of Each Factor (Biomarkers)	Potential Intervention
1. Persistent infection	Histological staining for NAPlr and plasmin activity [11]	Use of antimicrobial agents Removal of indwelling device
2. Genetic background of the host’s complement system [12]	Serum complement levels, histological deposition of complement components, genetic testing	Use of complement—regulating medications (in the future)
3. Tubulointerstitial changes	Interstitial staining for α-SMA	Not yet determined
4. Pre-existing renal histological damage due to comorbidities [4,13]	Histopathological evaluation	Adequate treatment for comorbidities, such as hypertension and DM

NAPlr: nephritis-associated plasmin receptor; α-SMA: alpha-smooth muscle actin; DM: diabetes mellitus. Numbers of related references are listed.

**Table 2 ijms-22-00905-t002:** Alpha-smooth muscle actin staining scores of renal biopsy tissues sampled repeatedly from 6 PSAGN patients and their characteristics.

	Patient Age (Years)	Acute Phase: Duration between Onset to First Biopsy (Days)	Convalescent Phase: Duration between Onset to Second Biopsy (Years)	Glomerular Staining Score		Interstitial Staining Score	
				First Biopsy	Second Biopsy	First Biopsy	Second Biopsy
Progression into CKD (*n* = 3)	39.3 ± 6.0	19.7 ± 9.9	3.0 ± 1.0	2.7 ± 1.2	0.7 ± 0.6	3.7 ± 0.6	3.0 ± 1.0
Cured (*n* = 3)	35.7 ± 7.8	23.3 ± 12.1	5.1 ± 4.5	3.0 ± 1.0	0.7 ± 0.6	2.7 ± 0.6	1.7 ± 0.6

PSAGN: poststreptococcal acute glomerulonephritis; CKD: chronic kidney disease. Data are expressed as the mean ± SD. Sections stained for α-SMA as described in Figure 4 were evaluated as follows. Glomerular staining intensities were semiquantitatively graded as 0 to 4, as previously reported by Alpers et al. [48], whereas interstitial staining intensities were graded semiquantitatively as 1 to 4 using the following scale: 1 = positive area < 10%; 2 = 10% to 20%; 3 = 20% to 30%; 4 ≥ 30%.

## Data Availability

The data presented in this study are available upon request.

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
