# Peer review of "Factors Affecting the Progression of Infection-Related Glomerulonephritis to Chronic Kidney Disease"

_ijms, 2021, doi:10.3390/ijms22020905_

Round 1

Reviewer 1 Report

This manuscript summarized the possible factors affecting the progression of infection-related glomerulonephritis, including persistent infection, genetic background of the host’s complement system, tubulointerstitial changes, and pre-existing histological damage. The paper was written well.

  • The Title does not read well. It should be corrected to “Factors affecting the progression of infection-related glomerulonephritis”。
  • This is a review paper, so it should avoid to cite the published reviews, especially in the section 2-5.
  • Section 5 with no references, is it the author's own opinion? There is no any literatures on pre-existing kidney injury in the progression of IRGN into CKD? Is there any relevant data to support it from the author’s lab?
  • Table 1, please provide the related references.
  • Figure 1-5 and Table 2, it looks like unpublished data, is it right? If so, please provide detail methods, and it can be provided as a supplementary materials. The images should be marked with a scale bar. If not, please provide the references.
  • Conclusion section should be simplified.

Author Response

Responses to the comments of Reviewer 1

We sincerely thank you for your kind and constructive comments and suggestions. Our point-by-point responses are shown below.

Point 1: The Title does not read well. It should be corrected to “Factors affecting the progression of infection-related glomerulonephritis”

Response: Following your advice, we reconsidered the title carefully. As the fundamental concept of this special issue is “transition of AKI to CKD”, we felt we should make a title that has some association with this concept. Therefore, we changed the title to “Factors affecting the progression of infection-related glomerulonephritis to chronic kidney disease”.

Point 2: This is a review paper, so it should avoid to cite the published reviews, especially in the section 2-5.

Response: Thank you very much for the important advice. In accordance with your suggestion, we avoided citing any review papers in section 2-5, except for one paper (reference no. 33, on page 4, line 145), which is our newest review.

Point 3: Section 5 with no references, is it the author's own opinion? There is no any literatures on pre-existing kidney injury in the progression of IRGN into CKD? Is there any relevant data to support it from the author’s lab?.

Response: Thank you very much for the important advice. We did not realize that we did not cite any papers in this section. We cited two papers in the revised version of the manuscript (ref. no. 4 and 51, page 9, line 295).

Point 4: Table 1, please provide the related references.

Response: Following your advice, we added the related references in Table 1.

Point 5: Figure 1-5 and Table 2, it looks like unpublished data, is it right? If so, please provide detail methods, and it can be provided as a supplementary materials. The images should be marked with a scale bar. If not, please provide the references.

Response: Figures 3 and 5 are published data, and the references for these pictures were cited in the original manuscript. For easier understanding, we simplified the way the reference numbers are indicated in the revised manuscript. On the other hand, Figures 2, 4, and 6 are unpublished data, and hence we added scale bars to the pictures and provided the references for the detailed methods or described them in the legends. Table 2 is also unpublished data, so we described the detailed methods in the legend of the revised manuscript.

Point 6: Conclusion section should be simplified.

Response: In accordance with the comment, we shortened the conclusion section of the revised manuscript.

Reviewer 2 Report

The review by Oda and Yoshizawa on how acute poststreptococcal glomerulonephritis may result in chronic renal disease is well written and comprehensive. The authors are experts in the field. The manuscript is worth publishing but I'd like the authors to assess two issues: 

Does the presence of certain antibodies such as anti-dsDNA or anti-proteinase 3 antibiodies result in a worse or better renal outcome? 

I would like to ask the authors to draft a comprehensive summarizing graphic figure to accompany the manuscript. 

Author Response

Responses to the comments of Reviewer 2

We sincerely thank you for your kind and constructive comments and suggestions. Our point-by-point responses are shown below.

Point 1: Does the presence of certain antibodies such as anti-dsDNA or anti-proteinase 3 antibiodies result in a worse or better renal outcome?

Response: Some reports have indicated the appearance of ANCA in patients with IRGN, particularly in those with glomerulonephritis associated with infective endocarditis. There have also been reports of lupus-like symptoms in patients with certain viral IRGN, such as parvovirus B19 and CMV IRGN, with the appearance of anti-nuclear antibodies and anti-dsDNA antibodies. Furthermore, the presence of anti-factor B autoantibodies has recently been reported in patients with acute post-infectious GN. However, these were mostly reports on a limited number of cases, and analyses of a large number of cases or prospective studies have not been reported to date. Furthermore, there appears to be a marked difference in the outcomes depending on the report regarding the association between appearance of autoantibodies and the prognosis of IRGN. Thus, the effect of the presence of autoantibodies on the prognosis of IRGN remains unclear, and is a matter of future investigation. These points were added to the revised manuscript as follows (page 2, lines 65−70, page 9, lines 300−310).

Some autoantibodies, such as the anti-neutrophil cytoplasmic antibody (ANCA), anti-nuclear antibody (ANA), anti-dsDNA antibody, and anti-factor B antibody have been reported to be detected in patients with IRGN [11-18]. Although these antibodies may contribute to the progression of IRGN, at present there is little data regarding their significance on the prognosis of IRGN, and as their involvement remains controversial, we touched on this point but did not list the antibodies as possible factors associated with IRGN.

ANCA, especially PR3-ANCA, has been reported to be detected in patients with IRGN, particularly that caused by infective endocarditis [11-14]. There have also been reports of lupus-like symptoms in patients with some viral IRGN, such as parvovirus B19 [15, 16] and CMV IRGN [17], with the appearance of ANA and anti-dsDNA antibodies. Furthermore, the presence of anti-factor B autoantibodies has recently been reported in patients with acute PIGN [18]. However, these were mostly reports of a small number of cases, and analyses of a large number of cases or prospective studies have not been reported to date. Furthermore, there appears to be a marked difference in outcomes depending on the report regarding the association between the appearance of autoantibodies and the prognosis of IRGN. Thus, the effect of the presence of autoantibodies on the prognosis of IRGN remains unclear, and is a matter of future investigation.

Point 2: I would like to ask the authors to draft a comprehensive summarizing graphic figure to accompany the manuscript.

Response: In accordance with the comment, we made a graphic figure summarizing the content of this review, as Figure 1 of the revised manuscript.

Reviewer 3 Report

      Authrors proposed their opinions to describe factors in infection-related glomerulonephritis to progress to CKD. Comments are as followed.

  1. Clinical nephrologists would have to know the NAPLr/plasmin acitivities from kidney bioipsies staining. They cannot benefit from this paper if they cannot access kidney biopsies. Therefore, in term of practice, this does not help more to physicians.
  2. Readers would like to read a review paper summarized from well-studied evidences. But obviously, much of the details of IRGN were not well-studied (both clinical and basic), even though authors had additionally described personal case series experiences.
  3. The text for "genetic background" is purely descriptive. Can authors provide published evidence on this? It is already a common sense that genetic heterogeneity must have played roles in diseases.

Author Response

Responses to the comments of Reviewer 3

We sincerely thank you for your kind and constructive comments and suggestions. Our point-by-point responses are shown below.

Point 1: Clinical nephrologists would have to know the NAPLr/plasmin acitivities from kidney bioipsies staining. They cannot benefit from this paper if they cannot access kidney biopsies. Therefore, in term of practice, this does not help more to physicians.

Response: We are nephrologists in Japan, and we actually directly evaluate kidney biopsy samples under the supervision of pathologists. Thus, in Japan, it is usual for nephrologists themselves to analyze renal biopsy tissues. Even if physicians in some other countries do not usually analyze renal biopsies themselves, it should be possible for them to consult the pathologists of their institution and examine the stained samples. Therefore, we believe our findings will be of great interest to nephrologists around the world.

Point 2: Readers would like to read a review paper summarized from well-studied evidences. But obviously, much of the details of IRGN were not well-studied (both clinical and basic), even though authors had additionally described personal case series experiences.

Response: We completely agree with the reviewer’s comment that little evidence exists regarding both clinical and experimental IRGN. Indeed, clinically, no prospective RCT have been performed to date, probably because of the rarity of this disease and because detailed diagnostic criteria have not yet been established. Furthermore, the establishment of reliable experimental animal model of IRGN has remained unsuccessful for many years, owing to the differences in the infectiveness of bacteria in different species. For example, GAS does not infect rodents.

However, these difficulties in obtaining evidences make reports of each IRGN cases all the more valuable. We cannot obtain information on IRGN without detailed evaluation of these valuable IRGN cases. Thus, we believe that at present, the accumulation of informative IRGN cases is crucial for understanding IRGN in general.

We discussed on this point briefly in the Introduction section of the revised manuscript as follows (page 2, lines 71−75).

Generally, clear evidence in this field is very scarce, because no large prospective clinical studies have been performed owing to the rarity of IRGN, and furthermore, a reliable animal model has not been established to date, partly owing to the differences in the infectiveness of pathogens among different species. Therefore, a substantial portion of this review is based on detailed evaluations of case reports.

Point 3: The text for "genetic background" is purely descriptive. Can authors provide published evidence on this? It is already a common sense that genetic heterogeneity must have played roles in diseases.

Response: We think the reviewer misunderstood what we meant by “genetic background”, and we apologize for our unclear explanation. We used the phrase specifically to mean “genetic abnormalities in the regulation system of the alternative complement pathway”. To avoid any misunderstanding, we have described this point more clearly in the revised manuscript as follows (page 6, lines 190−192).

In this previous report, 11 patients with so-called atypical PIGN were analyzed and 4 of them were found to have genetic abnormalities in the regulatory system of the alternative complement pathway (3 had a mutation in CFH, and 1 in CFHR5). [38]